# Revealing the assembly of filamentous proteins with scanning transmission electron microscopy

**Cristina Martinez-Torres**[1,2], **Federica Burla**[1], **Celine Alkemade**[1,2], **Gijsje H. Koenderink**[1,2]*

**1** Department of Living Matter, AMOLF, Amsterdam, the Netherlands, **2** Department of Bionanoscience, Kavli Institute of Nanoscience Delft, Faculty of Applied Sciences, Delft University of Technology, Delft, The Netherlands

* g.h.koenderink@tudelft.nl

**Data Availability Statement:** The dataset for STEM images is available in the 4TU.ResearchData website, https://doi.org/10.4121/uuid:e84d0eb1-23a0-40bb-b344-e0c61dfc2a22. The code used to apply the tracking algorithm and process the data

## Abstract

Filamentous proteins are responsible for the superior mechanical strength of our cells and tissues. The remarkable mechanical properties of protein filaments are tied to their complex molecular packing structure. However, since these filaments have widths of several to tens of nanometers, it has remained challenging to quantitatively probe their molecular mass density and three-dimensional packing order. Scanning transmission electron microscopy (STEM) is a powerful tool to perform simultaneous mass and morphology measurements on filamentous proteins at high resolution, but its applicability has been greatly limited by the lack of automated image processing methods. Here, we demonstrate a semi-automated tracking algorithm that is capable of analyzing the molecular packing density of intra- and extracellular protein filaments over a broad mass range from STEM images. We prove the wide applicability of the technique by analyzing the mass densities of two cytoskeletal proteins (actin and microtubules) and of the main protein in the extracellular matrix, collagen. The high-throughput and spatial resolution of our approach allow us to quantify the internal packing of these filaments and their polymorphism by correlating mass and morphology information. Moreover, we are able to identify periodic mass variations in collagen fibrils that reveal details of their axially ordered longitudinal self-assembly. STEM-based mass mapping coupled with our tracking algorithm is therefore a powerful technique in the characterization of a wide range of biological and synthetic filaments.

## Introduction

The main structural components of cells and tissues are scaffolds made of proteins that self-assemble into filaments. Cells are sustained by cytoskeletal filaments, whilst tissues are supported by an extracellular matrix composed predominantly of collagen fibrils. These protein scaffolds have unique material properties, combining a superior mechanical resistance to large deformations with the ability to dynamically adapt, grow and repair [1–3]. There is growing

is available in a GitHub repository (https://github.com/cristina-mt/fias).

**Funding:** This work was financially supported by the Netherlands Organization for Scientific Research (NWO) with a Topsector grant and a Veni fellowship (CMT). This work was further supported by the European Research Council (Synergy grant 609822). The contribution of FB and GHK is part of the Industrial Partnership Program Hybrid Soft Materials that is carried out under agreement between Unilever Research and Development B.V. and the NWO. The funders had no role in study design, data collection and analysis, decision to publish, or preparation of the manuscript.

**Competing interests:** The authors have declared that no competing interests exist.

evidence that the molecular packing structure of protein filaments is an important determinant of these unique material properties. Both cytoskeletal and extracellular filaments are supramolecular assemblies with a highly organized molecular structure dictated by specific noncovalent interactions between the constituent proteins. For a quantitative understanding of the relation between the mechanics and structure of these biopolymers, we need to be able to quantitatively characterize their internal molecular packing arrangement. This task is challenging as biopolymers are thin structures (with widths of 5–25 nm for cytoskeletal filaments and 10–500 nm for extracellular filaments) with small masses (1 kDa—1 MDa). One powerful tool to spatially resolve the mass density of supramolecular protein assemblies is scanning transmission electron microscopy. In this technique, a converging beam of electrons scans across a thin specimen and by collecting only those electrons that are scattered at very high angle, the so-called High Angle Angular Dark-Field Mode (HAADF) (Fig 1A), the resulting image intensity is directly proportional to the mass of the specimen[4,5]. The proportionality constant necessary to satisfy this relation can be obtained either by an extensive calibration of the electron microscope [6,7], or by imaging the protein of interest together with a reference specimen that serves as an internal mass calibration [5,8].

Although mass mapping with STEM has already been performed on multiple different systems [9–14], its applicability and the information content have been greatly limited by the approach used for data analysis. To obtain the mass per unit length of the filaments, typically a manual image analysis is performed: a box is drawn around the filament of interest, and the integrated intensity is computed inside this region, taking into account a correction for the background intensity surrounding the filament. This approach has several drawbacks. First, there is the limited statistics that can be reasonably reached by manually selecting the coordinates and/or width of the bounding box. Second, manual analysis requires user input in defining the bounding box, which could potentially introduce a considerable human bias in the results, particularly if the width of the filament is one of the variables to be analyzed. Finally, the discretization of the points along the filament axis could mask interesting features about spatial modulations in protein density along or across the filament axis. Many natural proteins form filaments that are highly ordered due to precise axially staggered self-assembly. The best-known example is the extracellular matrix protein collagen, a long rod-shaped molecule that assembles in a quarter-staggered manner to form filaments with a 67 nm axial periodicity referred to as the D-period [11,15].

Here, we report a semi-automated algorithm to retrieve the mass per length of filamentous structures and obtain high throughput information on the correlation between filament mass and morphology. Our method automatically tracks the filament skeleton and its borders via a modified canny edge detection with the wavelet transform [16–18], resulting in a robust tracking even in conditions of low image contrast for proteins of low density. The principle of this analysis is to smooth the image and compute its gradient, by convolving the image with an appropriate filter, such as the first derivative of a Gaussian function. This approach allows us to retrieve the distribution of mass per length and the widths for all fibers within a given image. Moreover, it enables us to quantify detailed structural features of fiber and network assembly, such as fiber internal packing, axial packing periodicity, fiber twisting and branching. The method we propose therefore opens up the way to quantitatively dissect the assembly of complex filamentous structures, from natural to synthetic fiber-forming systems.

## Results and discussion

The workflow of our algorithm is summarized in Fig 1B. The starting point is the unprocessed HAADF image (Fig 1C), where the intensity is minimal for the background (black) and

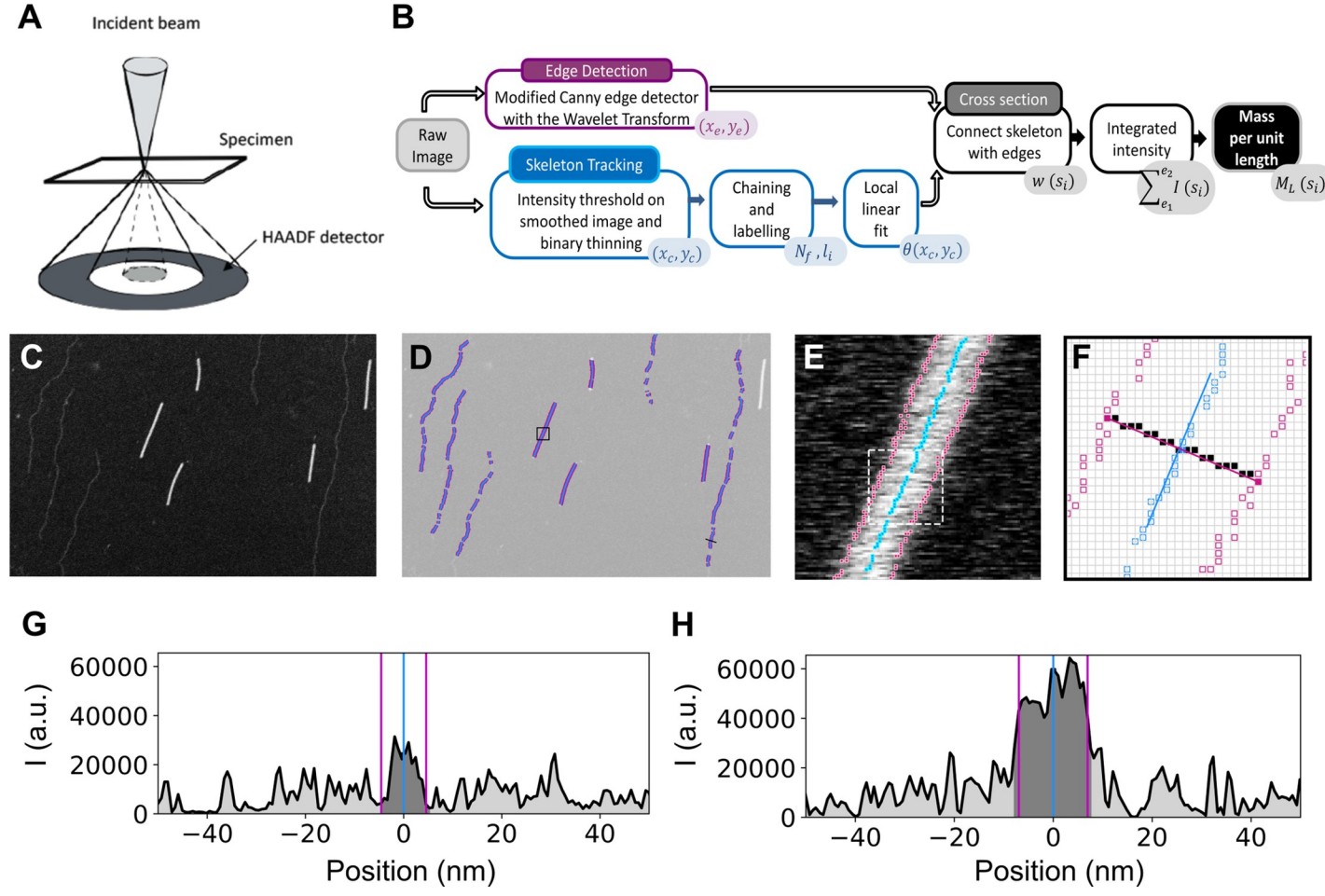

**Fig 1. Overview of the method for filament tracking and mass-per-length ($M_L$) computation.** (A) Scheme of the HAAFD mode, where the incident electrons are scattered by the specimen and collected at high angles with an angular detector. (B) Algorithm workflow showing the steps and the corresponding output (color-filled text boxes in the lower right corners). (C) Unprocessed image of TMV rods (bright short rods) and Fd bacteriophages (dim long filaments) serving as input in this example. (D) Tracked edges (magenta) and skeleton (blue) points, used to compute the $M_L$. Only the filament segments with a clean background and without intensity saturation are retained. (E) Image inside the black square shown in (D), with the tracked points overlaid. (F) Tracked edges and skeleton points inside the dashed white square shown in (E). The blue line shows the linear fit to the skeleton point in the center while the corresponding cross section is shown in magenta. The solid black squares represent the pixels in the filament cross section. (G) Intensity profile of the cross-section of the TMV rod in (F). The vertical lines show the detected edges and skeleton. Dark and light grey represent the integrated intensity for the filament and the background, respectively. (H) Same as (G) for an Fd filament indicated with a black line in (D).

maximal for the pixels containing the specimen that scatter the electrons (white). The method follows two independent tracking processes, one to detect the filament edges, and the other one to detect the filament main axis, or skeleton (Fig 1D–1E). The skeleton is detected by thinning a binary mask obtained by thresholding the intensity of the image (smoothed by a Gaussian filter). Next, neighboring points are connected into chains by a 8-neighbour rule and each chain is labeled with a different number to identify connected points. Short chains may be identified as noise and discarded at this point. The final step in tracking the filament skeleton performs a local linear fit around each point (Fig 1F), in order to obtain the local orientation of the filament and to retrieve transverse intensity profiles all along the axis of the filament. The fit range can be modified depending on the persistence length of the filament. In the edge tracking algorithm, we use the first derivative of a Gaussian as a kernel. When using this function in the wavelet transform, the result is a smoothed version of the 2D gradient of the image

[18]. Moreover, the transform gives an argument and a modulus for each point, that can be used as input for a Canny detection algorithm. At each image coordinate, a point is considered an edge if the wavelet transform modulus is a local maximum when compared to its neighboring pixels. The comparison is made with those pixels that follow the argument of the wavelet transform at that point. To identify 'true' edges, the points are chained together and a double hysteresis algorithm is applied to connect weak with strong edges [16]. Once the filament edges have been detected, they can be assigned to a point in the filament axis (skeleton), by looking for them in the cross-section profile (Fig 1F). The integrated density is then computed for the identified filament region, subtracting the intensity of the background on each side of the filament to correct for any scattering from the background and the support film (Fig 1G and 1H). If the constant of proportionality between the intensity and mass per length ($M_L$) is known, the integrated intensity can be converted to $M_L$ values. In this study, we use tobacco mosaic virus (TMV) rods for mass calibration, since these have a constant and well-known mass per length ratio [19]. However, depending on the morphology and mass range of the filaments to be studied, other well-calibrated structures can be used as reference, for example Fd bacteriophages [20], which appear as dim long filaments in Fig 1C and which likewise have a constant and well-known mass per length ratio (S1 Fig).

## Method validation on virus rods and cytoskeletal filaments

The cell owes its shape and mechanical properties to a filamentous scaffold known as the cytoskeleton. Two of the structural proteins forming the backbone of the cytoskeleton, actin and microtubules, are known to assemble into relatively simple and monodisperse structures. Therefore, we decided to use these proteins as test structures to validate our method. Actin filaments are helical polymers composed of two linear strands of globular actin subunits (Fig 2A, inset). With a width of only 8 nm and $M_L$ of 16 kDa/nm [9,21,22], an actin filament is one of the smallest structures we can measure given the accuracy of STEM mass mapping: in general an image resolution of 2–4 nm can be achieved depending on the sample preparation, while the mass range that can be detected is typically in the range of a few 10 kDa up to 100 MDa [4,23]. Fig 2A shows a HAADF image of an actin filament deposited on an EM grid covered with a 3 nm thick carbon film. Although it is possible to reach a higher resolution by increasing the magnification of the microscope, it is important to keep the electron dose per nm$^2$ to a minimum to prevent mass loss of the protein [5], and thus, an incorrect mass quantification. We optimized the imaging settings to enhance contrast, while maintaining the intensity of the reference TMV rods (lower right corner in Fig 2A) just below saturation. Our tracking method recovers an homogeneous distribution of $M_L$ values, with $<M_L> = 17.22 \pm 6.91$ kDa/nm and an average width $<w> = 15.77 \pm 1.82$ nm. The width of the filaments is larger than expected, which is likely explained by two factors. On the one hand, the spatial resolution is limited since 1 pixel typically corresponds to ~ 1 nm and the edge tracking method has a maximum accuracy of 2 nm in determining the width of a filament (±1 pixel on each border). We can thus expect an error on the order of 2–4 nm depending on the image resolution. On the other hand, the sample preparation can also change the width of the fibrils compared to the hydrated diameter in solution due to surface adsorption and drying. To prevent changes in the fibril width and, in extreme cases, possible structure collapse, it is preferred to use physical or chemical fixation, for example by freeze drying, vitrification [24], or glutaraldehyde fixation. Since we avoided chemical fixation in order not to change the $M_L$, the actin filaments likely flatten upon surface adsorption. As we were mainly interested in the mass of the filaments, we opted to use a simple method for sample drying, which does not require any specialized equipment. We next tested our method on microtubules, which form hollow sheets made of 12–15 linear

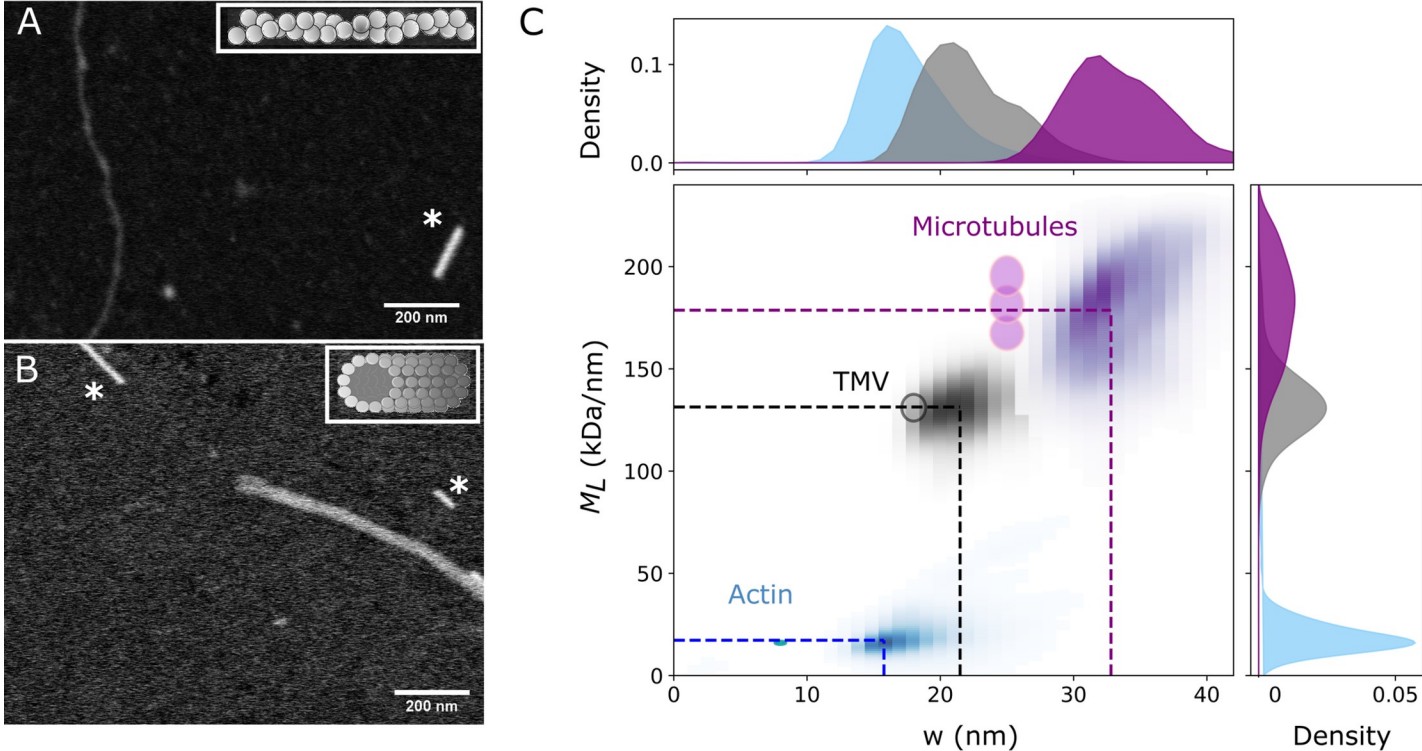

**Fig 2. Mass mapping of cytoskeletal proteins.** Unprocessed images of (A) an actin filament and (B) a microtubule. The short bright rods marked by an asterisk in both images are TMV rods used as reference. (C) Kernel density plot of the filament mass per length ($M_L$) and width (w) for data collected from multiple images of microtubules (purple), TMV (black) and actin (blue). The dotted lines show the mean fitted values, and the circles indicate the expected values from literature with a $\pm$ 10% spread. For microtubules, the three circles correspond to structures with 12, 13, and 14 protofilaments. $n$ = 8150 (actin), 2986 (microtubules), 5450 (TMV) segments.

protofilaments [25] (see inset of Fig 2B). Our tracking method recovers an $<M_L> =$ 178.7 $\pm$ 29.9 kDa/nm and a mean width $<w>$ = 32.8 $\pm$ 3.76 nm (Fig 2C), matching the values of native microtubules [26]. Microtubules exhibit less filament broadening than actin filaments, perhaps due to their larger rigidity. However, they exhibit a broader spread in $M_L$ values, consistent with the known variability in protofilament number [25,27]. Given our sample preparation (see Methods), we can expect microtubules with predominantly 13 and 14 protofilaments, with a respective $M_L$ of 181.5 kDa/nm and 195.4 kDa/nm, in agreement with our results. In addition, the microtubules we obtained have ends with a reduced $M_L$ (Fig 2B), likely because we did not chemically stabilize them.

## Structural analysis of more complex fibril-forming systems

The protein mass of filamentous structures is an essential parameter indicating the number of subunits present. Above we considered actin and microtubules, both proteins with a relatively simple and well-defined assembly process. In this case the filaments form a homogeneous population and the average $M_L$ and width values provide a sufficient description. However, many proteins of the extracellular matrix, like fibrin and collagen, form more complex and polymorphic supramolecular structures, and data averaging would obscure polymorphism within and between filaments. Since STEM provides concurrent information on the mass and shape of individual filaments, we propose that it is an attractive tool to study proteins that assemble into hierarchical structures. We demonstrate the potential of our filament tracking method in combination with STEM mass mapping by analyzing the assembly of collagen fibrils.

Collagen is the main structural element of the extracellular matrix of connective tissues, important for force transmission and load-bearing [28]. The morphology of the fibrils and the architecture of collagen fibril networks vary widely among tissues, providing tissue-specific mechanical functions. Reconstitution studies have shown that collagen assembly is regulated by many environmental factors, such as temperature, pH, and ionic strength [11,29,30]. A few studies have shown that STEM can be a useful tool to decipher the molecular pathway of collagen assembly [11] but a detailed analysis has been lacking due to limitations in image processing. To test the applicability of STEM, we polymerize collagen fibrils from bovine dermal collagen I and adsorb them to an EM grid. As shown in Fig 3A, the resulting fibrils are highly variable in width. The corresponding histograms of $M_L$ (Fig 3D) and width ($w$) (Fig 3E) show several peaks, indicative of multiple populations. Note that the high output of our method uniquely reveals this polymorphic behavior. In the image analyzed (Fig 3A), we recover the information from over 5000 transverse fibril segments. Segments where the background or the fibril were contaminated by salts or other unwanted structures, were manually filtered out from the dataset based on visual inspection.

The EM image reveals several morphological features known to be characteristic of collagen fibrils (Fig 3C): the fibrils are branched, bundled and twisted at large (micron) scales, and they have an axially periodic packing structure at the nanoscale [11,14,31]. The packing periodicity is clearly visible from the regular bands with alternating regions of high and low intensity (*i.e.* mass). Furthermore, the fibrils have tapered ends, consistent with prior observations both in tissue and reconstituted collagen [11]. The packing periodicity is often studied with atomic force microscopy [14] (AFM) which can provide morphological, but not mass information. When we image our EM sample by AFM (Fig 3B), we indeed observe the same qualitative features in the fibril topography as in the HAADF images of the fibril mass density (S2 Fig).

## Lateral assembly and fibril packing

The basic unit of a collagen fibril is the tropocollagen molecule, which consists of an uninterrupted triple helix of nearly 300 nm in length and 1.5 nm in diameter [32]. In addition, the molecule may contain two short extra telopeptides, N- and C-terminal domains of 11 to 26 residues, which account for only ~2% of the molecule but which are critical in the assembly of the fibrils. The absence of these telopeptides results in a loss of diameter uniformity and changes in the fibril assembly pathway [28,33]. Fibrils reconstituted from purified collagen usually exhibit high axial periodicity, but their degree of lateral packing order is still elusive. X-ray diffraction studies in tissue collagen indicate an almost crystalline molecular packing [29,31,34], but deviations from this model are often observed. The differences have been attributed to fibril curvature and twisting or to the coexistence of ordered and less-ordered domains along the fibril axis [35,36]. However, all of these explanations remain speculative, and are mostly supported by theoretical models of molecular packing, without clear experimental evidence.

To address the question of the degree of packing density in collagen fibrils, we have studied fibrils with the presence (telo-) or absence (atelo- collagen) of the telopeptides. Rather than collapsing all the data into simple $M_L$ and $w$ distributions (as done in Fig 3) we now focus on the evolution of $M_L$ with the fibril diameter ($w$). Fig 4B shows the resulting curve for telocollagen (in blue) and atelocollagen (in red), where all the data (7 images with 50 fibrils for telo-, 14 images with 85 fibrils for atelo- collagen) has been pooled together, binned and averaged over a width window of 5 nm, which is the lowest imaging resolution we have in our dataset. We found that, for thick fibrils ($w > 100$ nm), atelocollagen fibrils have considerably lower values of $M_L$ compared to telocollagen, indicating a looser molecular packing. Since the $M_L$ and $w$ of a single tropocollagen molecule is known ($m_o$ and $w_o$ respectively), we can estimate the

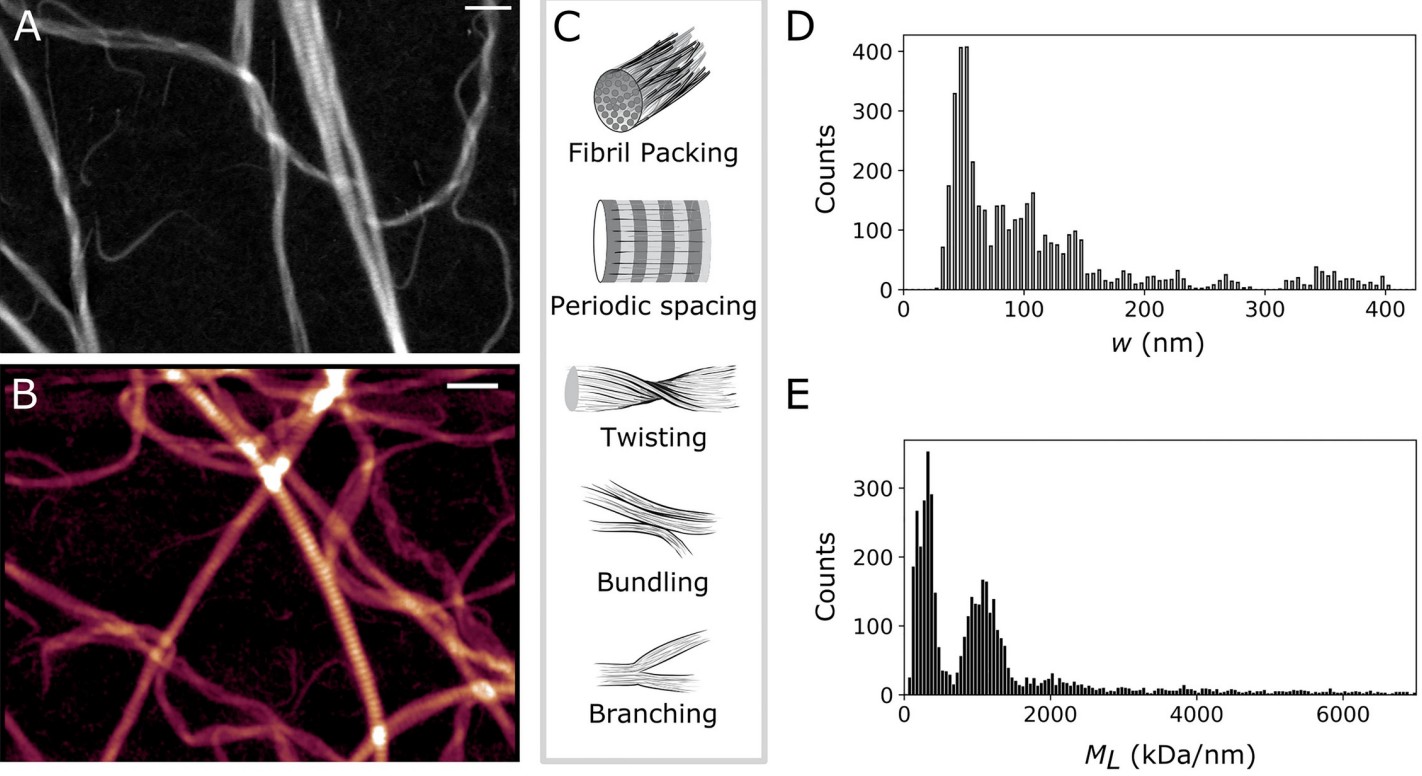

**Fig 3. Mass mapping of bovine type I collagen fibrils.** (A-B) Images of reconstituted atelocollagen fibrils adsorbed on an electron microscope grid. Images of the same sample were acquired using (A) HAADF mode with STEM, and (B) AFM contact mode in air (height color coded from 0 to 55 nm). Scale bar 500 nm. (C) Sketch of the different morphological features encountered in collagen fibrils, as seen in (A-B). (D-E) Distribution of the fibril width $w$ (D) and $M_L$ values (E) for the collagen fibrils in the image shown in (A). n = 5000 segments.

molecular packing density according to:

$$\frac{M_L}{m_o} = N_{TC} = \left(\frac{w}{w_o}\right)^{\alpha} \tag{1}$$

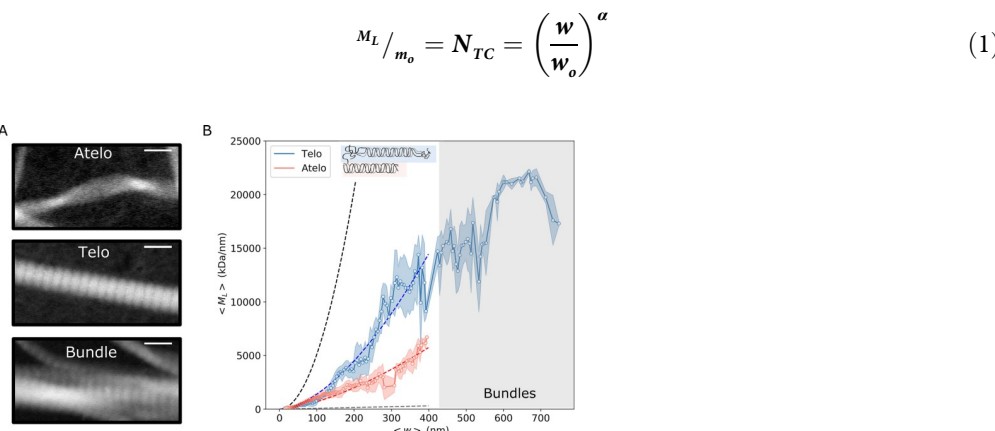

**Fig 4. Collagen fibril packing.** (A) HAADF images of representative fibrils from atelo- and telo-collagen. The bottom row shows an example of a bundle where multiple telocollagen fibrils join. Scale bar 200 nm. (B) Average $M_L$ values as a function of average fibril width, $w$, for telo- (blue) and atelo- (red) bovine collagen type I. The dotted lines show the expected $M_L$ values from Eq. (1) for $\alpha = 2$ (black) and $\alpha = 1$ (gray), and the fits with $\alpha = 1.687$ and $\alpha = 1.525$ for telo- (blue) and atelo-(red) collagen, respectively. Fibers in the grey-colored region are all bundles. The error bars represent the standard deviation. $n = 15695$ (telo-) and $n = 41730$ (atelo-collagen) segments.

where $N_{TC}$ is the number of tropocollagen molecules per cross section of the filament, and α is the power law describing the fibril packing density that can adopt values in the range of $1 \leq α \leq 2$, with α = 1 (dashed gray line in Fig 4B) corresponding to a quasi-bidimensional or flat structure, and α = 2 (dashed black line in Fig 4B) when the molecules uniformly occupy the interior of a cylindrical fibril. For intermediate values, $1 < α < 2$, we encounter a fractal structure, where the molecules are not as tightly packed. It is important to keep in mind that with STEM, as with any other technique where the biopolymer is interacting with a surface, some flattening of the fibrils can occur due to surface adsorption, which will also impact the packing density that we measure. Since we are mostly interested in the comparison between atelo- and telocollagen under the same conditions, the relative change of α values is still a meaningful parameter that is linked to the packing density in the fibrils. Using Eq. (1) to fit the data, we obtain α = 1.687 +- 0.003 for telocollagen, and α = 1.525 +- 0.003 for atelocollagen. This small difference in α translates in a large reduction in the fibril $M_L$ of up to 50% for the thicker fibrils (w ~ 400 nm) when the telopeptides are absent. The difference is particularly noticeable for widths larger than ~100 nm (S3 Fig). This could signify that telocollagen fibrils have a similar packing density in the core as atelocollagen fibrils, but a tighter packing in the periphery. Another interesting observation is that we observe two packing regimes for the telocollagen fibrils: the $M_L$ monotonically increases with width until ~400 nm, and thereafter increases more slowly. This threshold coincides with a clear change in morphology, from single fibrils for widths below ~400 nm to bundles above this threshold (grey-colored zone in Fig 4B).

## Longitudinal assembly and spatial features

A characteristic feature of the collagen fibrils is the so-called D-pattern or D-spacing, a periodic banded pattern appearing along the fibril axis. This axial periodicity is the result of the tropocollagen molecules aligning laterally in a quarter-staggered manner (Fig 5A), but details of the assembly process are still being studied with the aid of different theoretical models [15,29,33]. It has been proposed that the lateral molecular packing differs between the regions of the D-spacing, resulting in alternating regions of high packing density where the molecules overlap, and low packing density in the gap regions [29]. To test this hypothesis, we extracted the $M_L(w)$ values for the gap and overlap regions in a collagen fibril showing a clear D-pattern (Fig 5B). Starting from the semi-automated tracked points, we manually selected those sections that by visual inspection corresponded to either gap or overlap regions, giving a total of n = 547 and n = 733 cross-sections for ~80 gap and overlap regions, respectively. Fig 5C shows the resulting curves for the $M_L$ values averaged over 5 nm width bins. Fitting the data with Eq (1), we found α = 1.67 +- 0.002 for both the gap and overlap regions, suggesting that the alternating regions differ in their $M_L$ but not in packing density.

Our analysis also allows us to spatially map variations in $M_L$ and width along the fibril axis. Fig 5D shows the $M_L$ (black) and w (gray) profiles as a function of the fibril axis s, for the fibril segment enclosed by the asterisks in Fig 5B. Both the width and mass progressively increase as we move from the pointed end to the fiber middle. This behavior is reminiscent of tapered ends known as α-tips and β-tips that have been identified for fibrils assembling from procollagen. However, those tapered ends were reported to grow by only 17 and 113 molecules/D-period, respectively [37], whereas for the fibril shown in Fig 5D we find an increment of 248 ± 12 molecules/D-period, for the first 4 D-periods. This large difference in taper, together with the morphological appearance in the STEM image, suggests that the tapered ends we observe here are due to fibril splitting as a consequence of the sample preparation for EM imaging (see Methods) instead of a specific fibril growth mechanism [37]. The spatial map of the $M_L$ variation along the fibril furthermore enables us to infer a value for the average D-spacing of

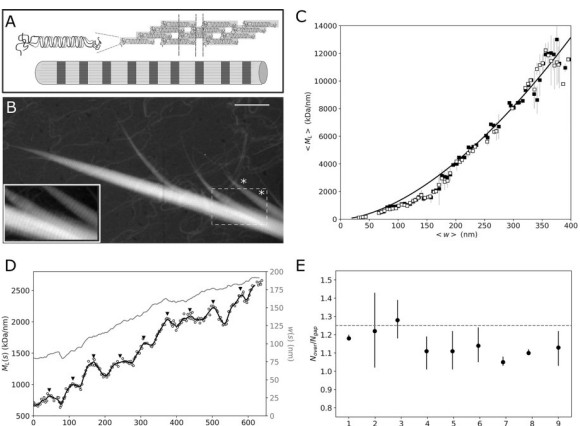

**Fig 5. Mapping axial order in a telocollagen fibril.** (A) Scheme of the quarter-staggered lateral assembly of collagen, resulting in a periodic pattern of overlap and gap regions. (B) STEM image of a single fibril from a telocollagen network. The inset is a magnification of the area in the white dashed square, where the contrast has been optimized to show the D-spacing pattern. Scale bar 1 μm. (C) Average $M_L$ values as a function of $w$ for the overlap (black squares) and gap (white squares) regions in the image shown in (B). The solid black line shows the fit to Eq. (1) with α = 1.67. (D) Axial mass profile for the fibril indicated with an asterisk in (B). The white circles show the raw $M_L$ data, and the black line the smoothed data. The corresponding width profile is shown in gray. The black upside triangles indicate the positions of the overlap regions $s_{over}$ (E) Ratio between the number of tropocollagen molecules per cross section in the overlap ($N_{over}$) and gap ($N_{gap}$) regions of the profile shown in (D). The x-axis shows the axial position of the overlap region ($s_{over}$), normalized by the mean D-spacing of the fibril, $<D>$ = 66.7 ± 5.4 nm.

$<D>$ = 66.7 ± 5.4 nm, consistent with prior reports based on negative stain EM and AFM [15,28,33]. The $M_L(s)$ profile shown in Fig 5D furthermore enables us to directly check the quarter-stagger assembly model (sketched in Fig 5A) by measuring the ratio in $M_L$ for the overlap and gap regions. We take for the positions of the overlap region, $s_{over}$, each of the local

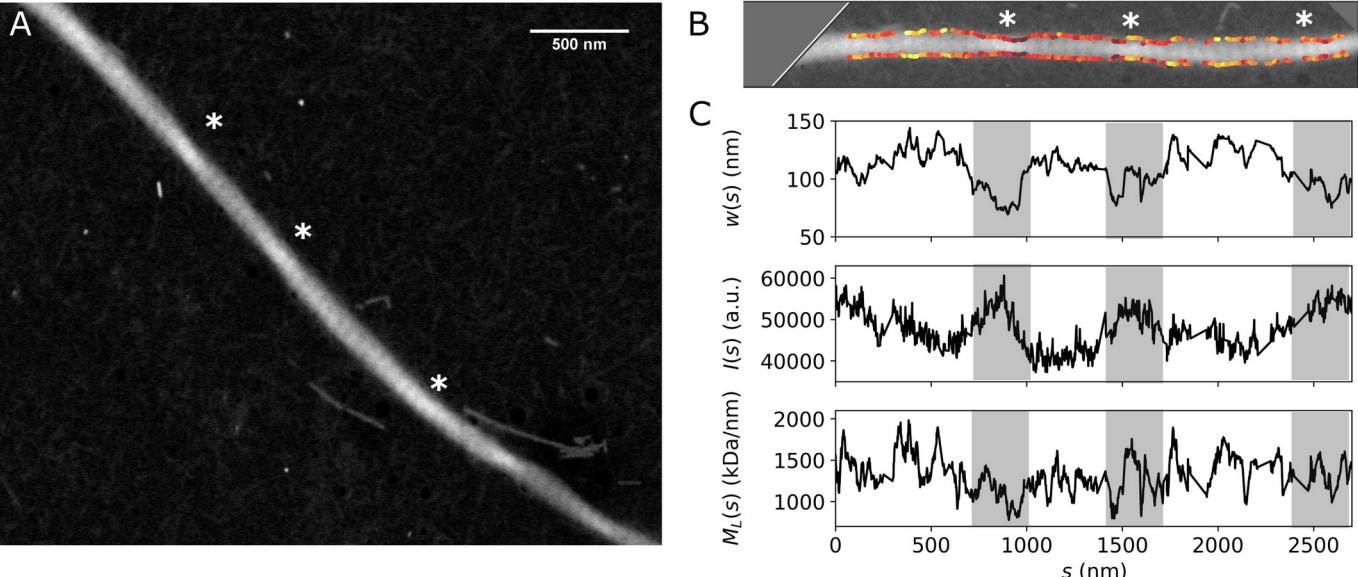

**Fig 6. Fibril twisting.** (A) STEM image of a single fibril from bovine telocollagen. The asterisks indicate the points where the intensity is enhanced and the morphology suggests the presence of fibril twists. (B) Cropped STEM image (shown in A), where the fibril has been rotated to be horizontal, and the contrast is optimized to show the twisting. The edge tracking is overlaid, with the $M_L(s)$ color coded in the same range shown in (C). (C) Axial profiles for fibril $M_L$, $w$, and intensity $I$. The gray rectangles serve as a visual aid to identify the regions where fibril twisting is observed in the image shown in (B).

maxima observed, and the number of tropocollagens per cross section in the overlap, $N_{over}$, as the value of $M_L(s_{over})/m_o$. For the gap regions, $N_{gap}$, we take the mean value from the local minima before and after $s_{over}$. Fig 5E shows the ratio $N_{over}/N_{gap}$, as a function of D-period. In all the overlap regions, we get a greater number of molecules compared to the gaps, regardless of the fibril width and/or distance from the tip. In this particular case, there is a mean ratio of $N_{over}/N_{gap} = 1.15 +- 0.006$, consistent with the theoretical ratio of 1.2 proposed in Ref. [29].

So far we have paid attention exclusively to the two main outputs of our tracking method for STEM mass mapping: the filament $M_L$ and width. However, we can gain additional information by correlating this information with secondary variables involved in the processing of the images. One of the fibril features that can be studied is the intensity, which is dependent, among others, on fibril twisting. Fig 6A shows the image of a collagen fibril that appears to be twisted at the points indicated by the white asterisks, where the intensity is higher and the fibril is thinner (Fig 6B and 6C). To make sure these variations in intensity are due to twist and not to differences in protein mass, we can correlate the mass profile $M_L(s)$ with the intensity profile $I(s)$ (Fig 6C). In this example the twisting points do not correlate with variations in fibril mass, suggesting that the $I(s)$ variations are indeed due to twisting. This phenomenon is reminiscent of prior reports of twisting in collagen fibrils [15], although it is unclear to what extent the interaction of the fibril with the surface plays a role here.

## Conclusions

We have developed a semi-automated algorithm to perform a detailed quantitative mass analysis of filamentous proteins from scanning transmission electron microscopy (STEM) images. Although different methods can be used for measuring the molecular packing structure of proteins, *e.g.*, small angle scattering using synchrotron radiation and optical turbidity measurements, they rely on bulk measurements and often involve assumptions about the network or protein structure to support the data analysis [38–40]. The spatial resolution and correlative image-mass analysis of our approach allows us to retrieve, in a label-free manner, information about the filament mass per length, molecular packing density, and about features related to the axial and longitudinal assembly of the protein. This information complements the structural information one can obtain by nanoscale infrared and enhanced Raman spectroscopy measurements, which provide spatially resolved information on the secondary structure of protein fibrils [41–44]. We have demonstrated the potential of automated STEM mass mapping by studying the structure of a wide range of biopolymers important in the cellular cytoskeleton and in the extracellular matrix. Our findings demonstrate that STEM has a wide mass range, from 17 kDa/nm for Fd bacteriophages and actin filaments to 20 MDa/nm for collagen bundles. However, we caution that attention must be paid to the thickness of the specimen [14], which should not exceed ~ 200 nm, and to possible sample deformation induced by the beam or the vacuum. We therefore recommend to validate the fibril thickness with an independent technique such as AFM, which, as we showed, can be performed on the same EM grid as used for STEM imaging.

A major advantage of the automated analysis over more traditional manual analysis approaches is the ease of obtaining high statistics where maps of the protein mass can be correlated with the local filament morphology. Our analysis of collagen fibrils shows that this approach is ideal for polymorphic systems of proteins that form filaments with a hierarchical, three-dimensional packing structure. There are many examples of such hierarchical fibril-forming proteins in nature where the hierarchy is essential for the mechanical and biological function. Axially periodic packing for instance ensures toughness and biological recognition in collagen [45,46] and high resilience in fibrin [47]. Although we have focused on the

description of ordered structures and molecules, mass mapping can also be used for filaments containing disordered regions, such as intermediate filaments [48] or fibrin [49], where information on the existence of unstructured sidearms can be inferred. Recently there is a growing effort in the chemistry community to design synthetic fibril-forming systems that mimic these features, based on peptides, DNA, or synthetic molecules [50–52]. STEM imaging coupled with our automated analysis provides an ideal tool to investigate the self-assembly processes of both natural and synthetic systems to aid in the design of filamentous (bio)materials.

# Materials and methods

## Actin sample preparation

The assembly of actin filaments was triggered by mixing globular (G-)actin (7.5 μM) purified from rabbit skeletal muscle [53] in an actin polymerization buffer solution containing 80 mM piperazine-N, N'-bis(2-ethanesulfonic acid) (PIPES), 4 mM $MgCl_2$, 75 mM KCl, and 1 mM EGTA, at pH 6.8. Actin was allowed to assemble for 1 hour at room temperature. Before deposition, the filaments were diluted to a final concentration of 1 μM in actin polymerization buffer. A 5 μL drop of the actin solution was deposited on electron microscopy grids, allowed to adsorb for 1 min at room temperature, rinsed 5 times with milliQ water, and blotted dry.

## Microtubule sample preparation

Lyophilized porcine brain tubulin (Cytoskeleton, Denver USA) was resuspended at 50–100 μM in MRB80 buffer (80 mM PIPES, 4 mM $MgCl_2$ and 1 mM EDTA, at pH 6.8), snap-frozen and stored at -80 ºC. Stabilized microtubule seeds were prepared using the slowly hydrolysable GTP analogue guanylyl-(α,β)-methylene-diphosphonate (GMPCPP), following the double cycle protocol as described in Ref. [54] and using 20 μM tubulin of which 12% was Rhodamine-labelled. To increase the average microtubule length, 20 μM tubulin was mixed with 1 μL of the GMPCPP seeds and 1.7 mM GTP in MRB80 buffer. Following incubation for 30–60 minutes at 37 ºC, a 5 μL drop of the microtubule solution was deposited on electron microscopy grids, allowed to adsorb for 1 min at room temperature, rinsed 5 times with milliQ water, and blotted dry.

## Collagen sample preparation

Collagen networks were reconstituted by polymerizing commercially available collagen extracted from bovine skin, with and without telopeptides–TeloCol and FibriCol respectively–(CellSystems, Germany). To trigger polymerization, the collagen solution was mixed on ice with 0.1 M sodium hydroxide (NaOH) and with phosphate buffer saline (PBS) to reach a final pH of 7.3–7.4, as measured by a pH meter. MilliQ water was added to top up the solution to the final volume. During the pH adjustment, samples were kept on ice to prevent premature polymerization of collagen. Fibril assembly was initiated by placing the samples in a closed container (comprised of the cap of a closed Eppendorf tube placed upside down) for at least 2 hours at 37 ºC. The collagen fibrils were transferred to the electron microscopy grid by peeling off the collagen gel drop surface with the grid. The sample was then rinsed 5 times with milliQ water and blotted dry.

## Scanning transmission electron microscopy (STEM)

Imaging was performed using the High-Angle Annular Dark-Field mode (HAADF) on a Verios 460 electron microscope (FEI, Hillsboro, OR, USA). The beam current was set to 100 pA and pixel dwell time to 3 μs, resulting in an electron dose of 400–1000 e⁻/nm² with a

convergence semi-angle of 2–10 mrad, depending on the specimen. To achieve molecular mass determination, the collected electron beam signal was calibrated using Tobacco Mosaic Virus (TMV, kindly provided by Jean-Luc Pellequer) as an internal mass calibration standard. In all samples, a drop of 2 μL of TMV solution (25 μg/mL in PBS) was added to the carbon-coated copper grids (Ted Pella, Redding, CA, USA) prior to sample deposition. TMV was allowed to adsorb onto the grid surface for 1 minute at room temperature, rinsed 3 times with miliQ water, and blotted dry. For imaging of Fd bacteriophages (kindly provided by Pavlik Lettinga), a 5 μL drop of a solution containing Fd rods in PBS was deposited on the grid containing TMV, allowed to adsorbed for 1 min, rinsed 5 times with milliQ water and blotted dry. For the actin and microtubules samples, the grids were first pretreated by placing them in a humidified chamber for 36 hours before use, to reduce the hydrophobicity of the carbon layer, and improve sample adsorption. After TMV adsorption and washing steps, a drop of 5 μL assembly buffer was added on the grids to keep them wet until sample deposition. Grids used for collagen samples did not require these extra steps in the preparation. For all samples, the imaging was performed within 2 hours following sample preparation.

### Atomic force microscopy imaging

After imaging the collagen samples with STEM, the grids were imaged by Atomic Force Microscopy (AFM) in contact mode in air, using a Veeco Dimension 3100 AFM. The scanning rate was set to 0.5 Hz, with a scan resolution of 512x512 pixels. Images were acquired with a SNL cantilever with a nominal spring constant of 0.06 N/m (Bruker).

### Filament tracking and mass determination

The unprocessed HAADF images were analyzed with custom mass mapping software written in Python. The software is freely available upon request and available to download from GitHub. Images are loaded and cropped to the area of interest. Then, the edge detection and tracking is performed by selecting an appropriate size for the kernel function, which will smooth and compute the gradient of the image; the threshold for the canny edge detection algorithm is chosen with the aid of visual validation of the tracked edges. The skeleton is tracked by choosing the size of the smoothing Gaussian kernel, and the threshold for converting to the binary image that will be skeletonized is validated with the aid of visual inspection. A local linear fit is performed around each skeleton point, and the intensity profile for each cross-section is retrieved. In between each step, undesired regions can be discarded from the analysis. At this stage, results are saved, and they can be loaded into the analysis module of the software, where mass mapping can be done, or analyzed with other custom tools based on Python. For mass mapping, a final visual inspection of the quality of the tracking is performed. The calibration constant for the conversion from image intensity to mass is computed by selecting the reference structures (TMV or Fd) in the image and comparing the integrated intensity values to the known mass per length. If there are no references in the image, but the calibration constant is known, it can be manually introduced. The resulting mass per length values, intensity, as well as the width, skeleton and edge coordinates for each cross-section in the tracked filaments are saved in a results file. All of the subsequent data processing (histograms, averaging, data fitting, plotting) was performed in Python, independently from our software.

### Supporting information

**S1 Fig. Distribution of mass per length values for Fd-bacteriopage filaments, used as an internal calibration standard.**
(PDF)

**S2 Fig. Atomic force microscopy of collagen fibrils.**
(PDF)

**S3 Fig. Collagen fibril packing regimes.**
(PDF)

**S4 Fig. Compilation of HAADF images from telocollagen and atelocollagen.**
(PDF)

**S1 File. Quick user guide of the developed GUI code for filament tracking and mass determination.**
(PDF)

## Acknowledgments

The authors thank A. Iyer, A. Szuba, V. Wollrab and A. Aufderhorst-Roberts for useful discussions; A. Lof for assistance with STEM and AFM imaging; and J-L. Pellequer and P. Lettinga for kindly gifting the TMV rods and Fd rods, respectively.

## Author Contributions

**Conceptualization:** Cristina Martinez-Torres, Gijsje H. Koenderink.

**Data curation:** Cristina Martinez-Torres.

**Formal analysis:** Cristina Martinez-Torres.

**Funding acquisition:** Cristina Martinez-Torres, Gijsje H. Koenderink.

**Investigation:** Cristina Martinez-Torres, Federica Burla, Celine Alkemade.

**Methodology:** Cristina Martinez-Torres, Federica Burla, Celine Alkemade.

**Software:** Cristina Martinez-Torres.

**Supervision:** Gijsje H. Koenderink.

**Validation:** Federica Burla.

**Visualization:** Cristina Martinez-Torres.

**Writing – original draft:** Cristina Martinez-Torres, Gijsje H. Koenderink.

**Writing – review & editing:** Cristina Martinez-Torres, Federica Burla, Celine Alkemade, Gijsje H. Koenderink.

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
