## [Decision Letter · Decision Letter 0]

24 Sep 2019

PONE-D-19-20842

Revealing the assembly of filamentous proteins with scanning transmission electron microscopy

PLOS ONE

Dear Dr. Koenderink,

Thank you for submitting your manuscript to PLOS ONE. After careful consideration, we feel that it has merit but does not fully meet PLOS ONE’s publication criteria as it currently stands. Therefore, we invite you to submit a revised version of the manuscript that addresses all the points raised during the review process.

Especially, describe the parameters of the electron beam carefully as well as the sample preparation method. Please also take care to discuss R2 concern about the thickness of the samples. 

We would appreciate receiving your revised manuscript by Nov 08 2019 11:59PM. To enhance the reproducibility of your results, we recommend that if applicable you deposit your laboratory protocols in protocols.io, where a protocol can be assigned its own identifier (DOI) such that it can be cited independently in the future. For instructions see: http://journals.plos.org/plosone/s/submission-guidelines#loc-laboratory-protocols

We look forward to receiving your revised manuscript.

Kind regards,

Etienne Dague, PhD

Academic Editor

PLOS ONE

Journal Requirements:

1. We note that you have stated that you will provide repository information for your data at acceptance. Should your manuscript be accepted for publication, we will hold it until you provide the relevant accession numbers or DOIs necessary to access your data. If you wish to make changes to your Data Availability statement, please describe these changes in your cover letter and we will update your Data Availability statement to reflect the information you provide.

Reviewers' comments:

Reviewer's Responses to Questions

**Comments to the Author**

1. Is the manuscript technically sound, and do the data support the conclusions?

Reviewer #1: Yes

Reviewer #2: Partly

2. Has the statistical analysis been performed appropriately and rigorously? 

Reviewer #1: Yes

Reviewer #2: Yes

3. Have the authors made all data underlying the findings in their manuscript fully available?

Reviewer #1: Yes

Reviewer #2: Yes

4. Is the manuscript presented in an intelligible fashion and written in standard English?

Reviewer #1: Yes

Reviewer #2: Yes

5. Review Comments to the Author

Reviewer #1: Could you please see my attached review, I have added all my comments to the Author in an attachment.

Reviewer #2: The work of Martinez-Torres et al. focuses on the use of STEM to probe the molecular mass and packing of proteins inside filaments. Even though the method has been developed two decades ago, the bibliography is rather thin, most probably because few groups were able to handle the method. In the past, the method was applied on single proteins mostly whereas Martinez-Torres et al. apply it on thicker specimens.

The bibliography covers the field, even though it would have been appreciated if more references were added, especially about other methods to characterize the structure of proteins such as AFM-IR or other methods based on interaction with light (lasers or synchrotron radiations).

The work is well presented, showing step by step the validation process, then the deeper and deeper characterisation of filaments.

The discussion lacks concerns about how the sample has been prepared. What about sample deformation induced by the beam, sample deformation inside vaccum, etc ? Also, the potentiel of the method on cryo-samples would be very interesting for the reader.

I have concerns regarding this work:

The first one is the lack of information about the parameters of the electron beam. The source current, the electron dose, the dwell time, the convergence semi-angle and the inner and outer collection angles are not defined. Without these numbers it is not possible to know the shape of the beam and what kind of electron/information was recorded on the HAADF detector.

The second concern is more important. STEM mass measurements using the HAADF detector can be safely performed on thin samples since electrons do not scatter more than once on thin samples. However, when thick samples are studied, HAADF signal is not fully representative of the sample (STEM tomography on thick samples is always performed in BF mode for the same reason). Furthermore, the convergence of the beam also has to be carefully set to very low values to avoid merging information from outer portions of the sample. Beam broadening is also another issue. Also, the results will be different if the beam was focused at the top or at the bottom of the filament. Based on the scale bars displayed in Fig. 4, the filaments can often measure over 200 nm, which is no longer in the thin sample range. In these conditions, the beam reaching the top of the filament and exiting the filament might have relatively large diameters, collecting blurry information from areas larger than the one where the beam is focused. The measurements made on thin samples might be valid (and the TMV rods used as reference is a good idea), however, for thicker filaments (telo and atelo) the values are probably wrong, likely explaining the difficulty to describe the values of the curves presented in Fig. 4.

The authors are asked to give the microscope settings used, and explain how they take into the various beam effects occuring when studying such thick samples.

6. PLOS authors have the option to publish the peer review history of their article (what does this mean?). If published, this will include your full peer review and any attached files.

Reviewer #1: No

Reviewer #2: No

---

## [Author Response · Author response to Decision Letter 0]

5 Nov 2019

Dear Dr. Dague,

We thank you for forwarding the reviews of our manuscript PONE-D-19-20842 entitled “Revealing the assembly of filamentous proteins with scanning transmission electron microscopy”. Both reviewers made several valuable suggestions to help us strengthen the manuscript, which we have addressed in the revised manuscript that we hereby submit. 

Briefly, we included details on the parameters of the electron beam and the choice of sample preparation method and we carefully discuss the limitations posed by larger (>200 nm) sample thickness (prompted by Reviewer 2). We have furthermore written a ‘quick user guide’ to explain the basic functionality of the software and provide guidance on troubleshooting, in order to encourage other researchers to apply our software (prompted by Reviewer 1). In the same spirit, we now provide a test image for the user to try and get familiarized with the software in the supplementary information and in the GitHub repository hosting the code.

We uploaded a marked-up copy of our manuscript that highlights the changes made to the original version (labeled 'Revised Manuscript with Track Changes'), an unmarked version of our revised paper without tracked changes (labeled 'Manuscript') and an itemized rebuttal letter labeled 'Response to Reviewers'.

Yours sincerely, Gijsje Koenderink

---

## [Decision Letter · Decision Letter 1]

25 Nov 2019

Revealing the assembly of filamentous proteins with scanning transmission electron microscopy

PONE-D-19-20842R1

Dear Dr. Koenderink,

We are pleased to inform you that your manuscript has been judged scientifically suitable for publication and will be formally accepted for publication once it complies with all outstanding technical requirements.

With kind regards,

Etienne Dague, PhD

Academic Editor

PLOS ONE

Additional Editor Comments (optional):

Reviewers' comments:

Reviewer's Responses to Questions

**Comments to the Author**

1. If the authors have adequately addressed your comments raised in a previous round of review and you feel that this manuscript is now acceptable for publication, you may indicate that here to bypass the “Comments to the Author” section, enter your conflict of interest statement in the “Confidential to Editor” section, and submit your "Accept" recommendation.

Reviewer #1: All comments have been addressed

Reviewer #2: All comments have been addressed

2. Is the manuscript technically sound, and do the data support the conclusions?

Reviewer #1: Yes

Reviewer #2: Yes

3. Has the statistical analysis been performed appropriately and rigorously? 

Reviewer #1: Yes

Reviewer #2: Yes

4. Have the authors made all data underlying the findings in their manuscript fully available?

Reviewer #1: Yes

Reviewer #2: Yes

5. Is the manuscript presented in an intelligible fashion and written in standard English?

Reviewer #1: Yes

Reviewer #2: Yes

6. Review Comments to the Author

Reviewer #1: Please correct spelling of the new reference, which should read "Krzyzanek" not " Krxyzanek". The authors have responded satisfactorily to the reviewers' comments.

Reviewer #2: I would like to thank the authors for providing a revised version of the manuscript with new/edited text in red, helping the revision process.

The message about thick samples has been correctly tuned down in the conclusion.

The various modifications greatly improve the value of the manuscrit (especially the presence of a user guide).

Unfortunately, the revision time is too short to allow testing the software, but the effort in providing all material is aknowledged.

That is why I know recommend publication of the manuscript.

7. PLOS authors have the option to publish the peer review history of their article (what does this mean?). If published, this will include your full peer review and any attached files.

Reviewer #1: No

Reviewer #2: No

---

## [Editor Report · Acceptance letter]

12 Dec 2019

PONE-D-19-20842R1 

Revealing the assembly of filamentous proteins with scanning transmission electron microscopy 

Dear Dr. Koenderink:

I am pleased to inform you that your manuscript has been deemed suitable for publication in PLOS ONE. Congratulations! Your manuscript is now with our production department. 

With kind regards,

on behalf of

Dr. Etienne Dague 

Academic Editor

PLOS ONE